# Bone marrow derived stromal cells from myelodysplastic syndromes are altered but not clonally mutated in vivo

Johann-Christoph Jann [1], Maximilian Mossner[1], Vladimir Riabov[1], Eva Altrock[1], Nanni Schmitt [1], Johanna Flach[1], Qingyu Xu[1], Verena Nowak[1], Julia Obländer[1], Iris Palme[1], Nadine Weimer[1], Alexander Streuer[1], Ahmed Jawhar[2], Ali Darwich [2], Mohammad Jawhar[1], Georgia Metzgeroth[1], Florian Nolte[1], Wolf-Karsten Hofmann[1] & Daniel Nowak [1✉]

The bone marrow (BM) stroma in myeloid neoplasms is altered and it is hypothesized that this cell compartment may also harbor clonal somatically acquired mutations. By exome sequencing of in vitro expanded mesenchymal stromal cells (MSCs) from n = 98 patients with myelodysplastic syndrome (MDS) and n = 28 healthy controls we show that these cells accumulate recurrent mutations in genes such as *ZFX* (n = 8/98), *RANK* (n = 5/98), and others. MDS derived MSCs display higher mutational burdens, increased replicative stress, senescence, inflammatory gene expression, and distinct mutational signatures as compared to healthy MSCs. However, validation experiments in serial culture passages, chronological BM aspirations and backtracking of high confidence mutations by re-sequencing primary sorted MDS MSCs indicate that the discovered mutations are secondary to in vitro expansion but not present in primary BM. Thus, we here report that there is no evidence for clonal mutations in the BM stroma of MDS patients.

[1] Department of Hematology and Oncology, Medical Faculty Mannheim of the Heidelberg University, Mannheim, Germany. [2] Department of Orthopedic Surgery, Medical Faculty Mannheim of the Heidelberg University, Mannheim, Germany. ✉email: Daniel.nowak@medma.uni-heidelberg.de

While there has been substantial progress in the identification of the molecular landscape and pathogenesis of myelodysplastic syndrome (MDS)[1–3], increasing evidence has also suggested that MDS may not only be a disease of hematopoiesis but also of the surrounding bone marrow (BM) microenvironment, also termed BM niche or stroma[4,5]. An active role of the BM niche in MDS but also other myeloid neoplasms such as MPN or AML has been demonstrated in several experimental approaches[6–9]. In patient-derived xenograft experiments, we have previously shown that primary MDS samples were dependent on the interaction with BM-derived mesenchymal stroma cells (MSCs) for their propagation[10]. This led to the hypothesis that MDS hematopoietic cells may re-educate their BM niche to create a supportive environment for their preferential growth. In support of this hypothesis, several studies have described aberrant gene expression and epigenetic profiles of MDS-derived MSCs[7,10–14]. The selective introduction of mutations to non-hematopoietic cellular compartments of the BM niche such as osteolineage cells[15–17] was able to disrupt hematopoiesis, induce MDS-like diseases, and control leukemic propagation[18]. Inflammatory programs mediated by S100A family proteins[7] and Toll-like receptor 4 (TLR4)[19] in mesenchymal precursors have further been shown to drive genotoxic stress in hematopoietic stem- and progenitor cells, proposing a mechanism of accumulation of genetic mutations in hematopoietic cells. Moreover, especially aging-related changes in the BM stroma are hypothesized to confer growth advantages for putatively pre-malignant hematopoietic subclones carrying typical mutations of early clonal hematopoiesis of indeterminate potential (CHIP)[20] such as DNMT3A[21].

An open question in the field is, therefore, whether there may also be acquired mutations in the BM stroma compartment of MDS patients, which act as causative or contributing pathogenic factors in the disease. Previous studies have identified chromosomal aberrations in ex vivo expanded MSC cultures from MDS and AML patients[22–25]. However, little validation has been performed so far to address whether such molecular lesions were not merely clonal outgrowths resulting from the strenuous and massively expansive cell culture procedures.

We and others have previously used MDS-MSCs as highly purified germline controls for genomic profiling of the MDS hematopoietic clones[10,26].

In this work, we present comprehensive data from whole-exome sequencing of in vitro expanded MSCs from $n = 98$ MDS patients and $n = 28$ healthy age-matched individuals to interrogate the question of acquired mutations in MSCs from MDS patients. While recurrent mutations can be detected in ex-vivo expanded MDS MSCs, these are not reproducible in serial BM aspirations from the same patients and do not remain stable in serial culture passages. Ultimately, high confidence mutations detectable in ex vivo MDS MSC cultures cannot be backtracked by targeted re-sequencing of primary sorted BM MSCs from the same patients. While MDS MSCs are biologically and functionally altered in comparison to healthy MSCs, our data indicate that there is no evidence for clonal somatically acquired mutations in non-hematopoietic BM stromal cells in MDS patients in vivo.

## Results

### In vitro cultured MDS MSCs carry more recurrently acquired mutations than healthy MSCs. Within the scope of previous molecular studies on MDS[10,26], we have used in vitro cultured MSCs as germline controls for exome sequencing of paired hematopoietic cell fractions. In order to address the question of whether MDS patient-derived MSCs also carry acquired molecular lesions, we took advantage of this data by performing a reversed bioinformatics approach using the corresponding hematopoietic fraction as a germline control for the MSC exome sequencing data. With this approach, we analyzed whole-exome sequencing results of MSCs derived from a total of $n = 98$ MDS patients (Fig. 1a; Table 1). The purity of the expanded MSCs was routinely confirmed to be completely depleted of residual hematopoietic cells (CD45 and lineage markers CD2, CD3, CD4, CD7, CD8, CD10, CD11b, CD14, CD19, CD20, CD56, and CD235a), and largely negative for endothelial CD31. Both healthy control- and MDS-MSCs showed positive expression for stroma cell markers such as CD146, CD271, CD105, CD73, and CD90 that were not significantly differentially expressed between these two groups (Supplemental Fig. 1a, b).

The initial whole-exome sequencing dataset of 98 MSC samples produced a predominantly low-level mutational spectrum with a median variant allele frequency (VAF) of 5.3% in all called mutations (Fig. 1b). By additional filtering for an exome-sequencing typical VAF cutoff of >10%, we identified a total of $n = 9857$ somatically acquired mutations (9419 SNVs and 438 Indels) in the MDS MSCs (Fig. 1c). Of note, as MDS-derived hematopoietic cells can carry acquired copy number changes, this was carefully accounted for by excluding affected genomic regions from analysis in the respective samples on an individual basis. In total, this data adjustment led to a marginal loss of region coverage of a mean of 1% per sample (Supplemental Data 2). In order to identify high confidence candidate mutations, we plotted the number of mutations against the estimated mutational significance determined by the MutSigCV tool[27] (Fig. 1b, c, Supplemental Data 3). Based on biological parameters such as functional genomic location, protein damage prediction, and others, this tool estimates the chance that a gene is mutated more often than expected by chance after adjusting for background mutational processes. Using this approach, we aimed to identify all genes with a combination of high recurrence, high MutSigCV scores, and high VAFs. Genes identified with a high recurrence but low MutSigCV significance values were likely to be biological or technical artifacts either due to the large size of the coding regions of these genes or local genomic susceptibility to mutation[27]. Among such low significance genes were TTN ($n = 19$, 19%), LRP2 ($n = 10$, 11%), or MUC16 ($n = 10$, 10%), of which recurrence increased further up to $n = 60$ when removing the 10% VAF cutoff.

The most preeminent recurrently mutated genes were zinc-finger protein X-linked (ZFX) ($n = 8$, 8% $p = 0.0008$), and RANK (Tumor necrosis factor receptor superfamily member 11A, TNFRSF11A, $n = 5$, 5%, $p = 0.02$; Fig. 1d, e, Supplemental Data 3). All mutations in the ZFX gene were either stop-gain or frameshift mutations with SIFT prediction of deleterious amino acid substitutions. ZFX is X-chromosome-linked and mutations were exclusively found in male patients. All five mutations discovered in RANK were predicted with deleterious impact to protein conformation (4/5) or affected a splice site, suggesting loss of function mutations (Fig. 1e).

To evaluate whether the acquisition of such mutations in cultured BM MSCs of MDS patients was possibly associated with myeloid neoplasia we also performed exome-sequencing on MSC cultures in a control cohort of $n = 28$ healthy age-matched individuals with their paired hematopoietic BM. In this dataset, we were also able to detect high confidence mutations. When applying 10% VAF cutoff we detected overall fewer mutations in this healthy control group as compared to the MDS cohort (median $n = 36$ per sample in MDS versus $n = 26$ per sample in healthy, $p = 0.0005$) (Fig. 1f). Notably, this difference was not accountable to the differential age distribution of the cohorts (Fig. 1g, Table 1). Moreover, in the unfiltered data sets, the overall VAF of detected variants was lower in healthy cells as compared

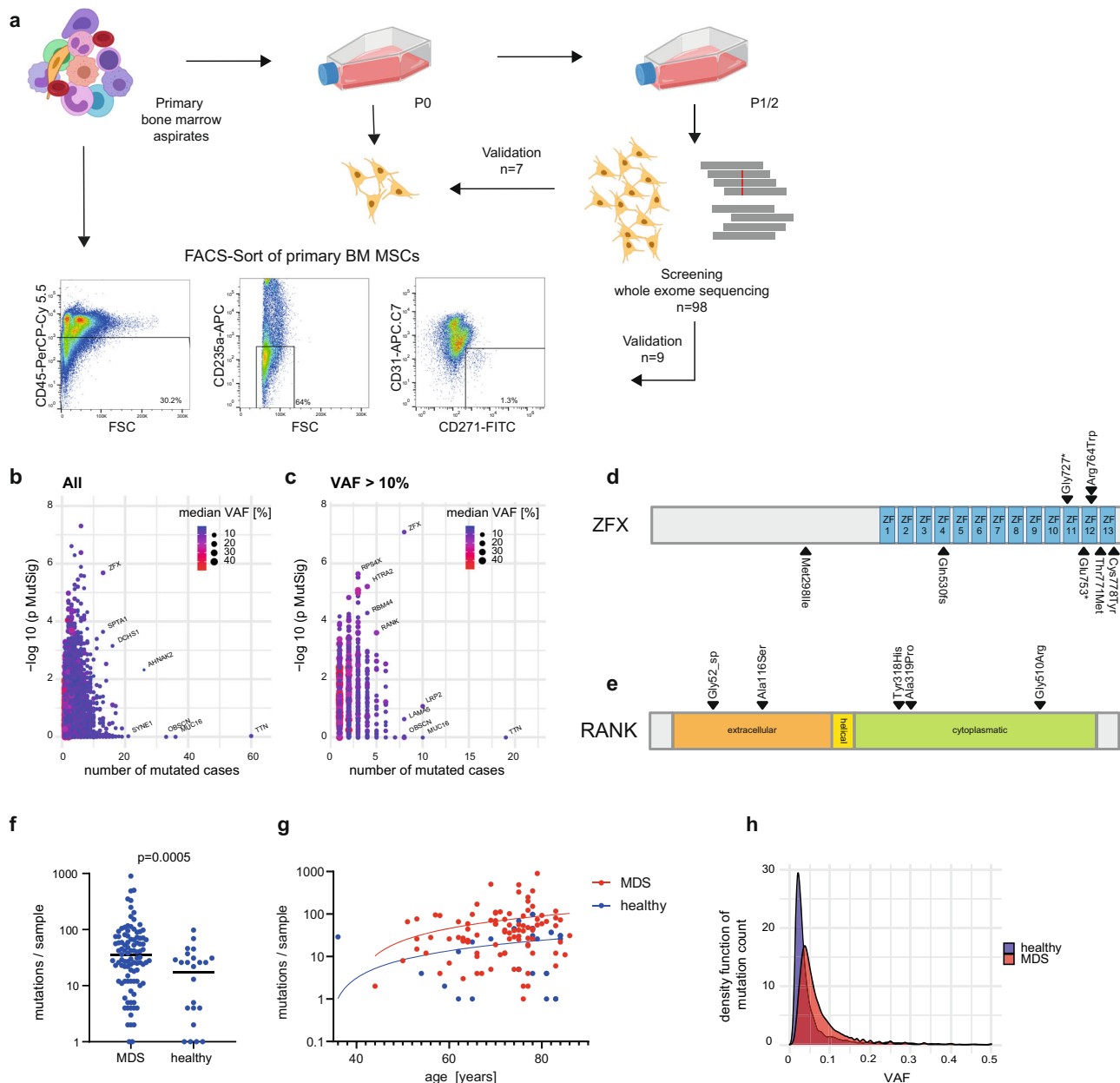

**Fig. 1 Recurrent mutations in MDS-derived MSCs. a** Scheme of the experimental workflow. P0/P1/P2 = number of culture passages, FSC forward scatter. **b** Summary of all detected recurrently mutated genes in MSCs of 98 MDS patients ($n = 34,855$ variants). The number of recurrences is displayed on the $x$-axis while the estimated biological significance calculated by MutSigCV is displayed on the $y$-axis (two-sided MutSigCV $p$-value, transformed as −log10). Exposed genes were enumerated, dot size and color represent the median VAF (variant allele frequency) per gene. **c** Data from (**b**) curated with a >10% VAF cutoff ($n = 9857$ variants). **d**, **e** Visualization of the recurrent mutations in the *ZFX* and *RANK* genes and their location relative to known domains derived from UniProt. **f** Number of mutations per sample for mutations with VAF > 10% (two-sided Mann–Whitey test) in MDS versus healthy controls. Data are presented as median and individual values. **g** Number of mutations per sample plotted against the age of the sample donor. **h** Density histogram of the total number of mutations per sample, MDS $n = 34,855$ mutations, healthy $n = 4506$ mutations. Source data are provided as a Source Data file.

to MDS (median VAF 5.3% for MDS versus 2.0% for healthy, $p = 2.2e−16$, Fig. 1h). The most frequently mutated genes in the healthy cohort were *TTN* and *SYNE1* (in 6 and 5/28 cases, 21 and 17%). Also, other genes detected in the MDS cohort such as *MUC16* or *LPR2* were frequently mutated in the healthy cohort. All recurrently mutated genes in the healthy control cohort were summarized in Supplemental Data 4. Of note, the most significantly mutated genes from the MDS cohort (*RANK* or *ZFX*) were not found to be mutated in the healthy control group.

Since most detected mutations in MSCs had low-level VAFs we performed several lines of validation experiments to confirm the

robustness of the above-described results. Firstly, we performed custom amplicon-based targeted deep re-sequencing (TDS) of a total of $n = 120$ mutations in $n = 117$ genes in $n = 12$ samples (Supplemental Data 5). Thereby, we confirmed a strong correlation of VAFs between exome sequencing and TDS (Fig. 2a, $r = 0.85$, $p < 0.0001$). Notably, all mutations in *ZFX* and *RANK* were confirmed by TDS. Next, we exemplarily validated the strategy of exome sequencing MSCs versus BM MNCs as germline control compared to another potential germline specimen such as sorted CD3-positive T-cells from peripheral blood or buccal swab DNA from the same patient (Fig. 2b). We observed a

**Table 1 Patient and healthy donor characteristics.**

| | MDS cohort $n = 98$ | Healthy control $n = 28$ | *p*-value |
|---|---|---|---|
| *Population data* | | | |
| Age, mean (range) | 70.8 (44-86) | 71.4 (36-84) | 0.78* |
| Sex | | | |
| Male | 65 | 14 | 0.13# |
| Female | 33 | 14 | |
| *WHO 2016* | | | |
| aCML | 2 | | |
| CMML | 3 | | |
| del5q-MDS | 15 | | |
| MDS-MLD | 33 | | |
| MDS/MPN-U | 2 | | |
| MDS-EB1 | 12 | | |
| MDS-EB2 | 12 | | |
| MDS-MLD-RS | 9 | | |
| MDS-MLD-RS-T | 1 | | |
| MDS-U | 3 | | |
| sAML | 4 | | |
| tMN | 2 | | |
| *Bone marrow cytogenetics* | | | |
| Normal | 52 | | |
| Complex aberrant | 14 | | |
| del5q | 17 | | |
| del7q | 1 | | |
| del9 | 1 | | |
| trisomy 8 | 4 | | |
| Other | 4 | | |
| n.a. | 5 | | |
| *Treatment prior to MSC sampling* | | | |
| ESA | 14 | | |
| Lenalidomide | 14 | | |
| Post cytotoxic therapy | 3 | | |
| Post HMA | 16 | | |
| No treatment | 51 | | |

*ESA* erythropoiesis stimulating agents, *HMA* hypomethylating agents.
*Two-sided students *t* test.
#Two-sided Fisher exact test.

strong overlap for the called mutations with only 1 call (2%) being exclusive to the BM versus MSC comparison. Finally, to confirm the clonality of the detected mutations we genotyped CFU-Fs derived from pre-expanded MSCs from P1 in $n = 4$ patient cases in a total of 23 colonies. This functional clonality assay likewise confirmed the previously detected mutations in corresponding ratios (Fig. 2c, d).

**MDS derived MSCs are functionally and molecularly altered in comparison to healthy MSCs.** The preceding data showed that MDS-derived MSCs seemed to be more susceptible to acquire mutations when in culture as compared MSCs from healthy individuals. We, therefore, performed functional analyses on MSCs from these cohorts to elucidate possible explanations for this higher rate of mutation acquisition. Since genotoxic stress in the HSC-stromal cell interaction induced by inflammatory phenotypes of MSCs was described to play a central role in the pathogenesis of human myeloid disease[7], we assessed these phenotypes in our MSC cultures. Thereby, we found markers for genotoxic stress such as frequency of γH2AX foci[28] or phosphorylation of replication protein A (RPA)[29] to be significantly increased in MDS-derived MSCs as compared to healthy MSCs (Fig. 3a–c). γH2AX staining correlated with the frequency of mutations in the interrogated MDS samples (Fig. 3d). Moreover, as expected from previous studies[30–32], MDS-derived MSCs presented

with increased senescence markers such as overall reduced telomere lengths and increased β-galactosidase levels as compared to healthy MSCs (Fig. 3e–h). Senescence is frequently coupled with an altered secretory phenotype referred to as senescence-associated secretory phenotype (SASP)[33]. We, therefore, analyzed RNA sequencing data from an extended group of $n = 8$ MDS MSC samples and $n = 6$ healthy MSC samples (partly previously published[10], EGAS00001000716) and observed higher expression of inflammatory markers of SASP in MDS MSCs (Fig. 3i).

These molecular perturbations in the bone marrow of myeloid neoplasms are frequently associated with an inflammatory phenotype in the bone marrow of myeloid neoplasms[4,34]. We could confirm this in MSCs from our cohorts by asserting increased gene expression levels of inflammatory genes such as IL-6 in our MDS MSCs versus healthy MSCs in a group of $n = 32$ MDS and 19 healthy cases (Fig. 3j–l).

Next, we asked whether these broadly observed phenotypic changes of increased DNA damage and increase of SASP profiles were possibly linked to differential mechanisms of mutation acquisition in MSCs and therefore determined the predominating COSMIC mutational signatures according to Alexandrov et al.[35] (Fig. 3m). Such mutational signatures resemble characteristic combinations of mutation types arising from various mutagenesis processes such as DNA replication infidelity, genotoxic exposures, or defective DNA repair. Most frequently, we found COSMIC signature 1, corresponding to spontaneous deamination of 5-methylcytosine[35] in 45/98 (46%) of MDS cases (Fig. 3m). Other frequently detected signatures were 24 and 29 in 40/98 (40%) of MDS cases, respectively.

While the signature data was markedly heterogeneous with almost all COSMIC signatures being present in at least one sample, there was a clear separation of healthy MSCs from MDS MSCs due to a significant over-representation of signatures 6, 12, and 15 in the healthy cells as compared to MDS derived MSCs ($p < 0.001$, Fig. 3m, n). This enabled a clear prediction of sample origin by the extracted mutational signatures (Fig. 3o), (AUC = 0.96). Collectively these data suggested that MDS MSCs were molecularly and functionally altered and acquired heterogeneous, yet distinct mutational profiles as compared to healthy MSCs.

Finally, since RANK is one of the central regulators of bone morphogenesis[36], we analyzed, whether *RANK* mutated MSCs possibly had altered differentiation dynamics. Upon in vitro osteogenic differentiation of $n = 2$ RANK mutated cases versus $n = 2$ RANK wild-type MDS cases we observed a higher osteogenic propensity of RANK mutant MSCs as compared to RANK wild-type MDS MSCs (Fig. 3p, q). These data suggest proof of principle that in vitro cultured MSCs may underlie functional differences in dependency of acquired mutations during culture expansion. Nevertheless, this generally also has to be evaluated in the context of the biological significance of recurrent mutations as performed above with the MutSigCV tool. Therefore, we also compared the mutational profiles with the RNA sequencing data from cultured MSCs. Table 2 lists the mean gene expression by fragments per kilobase of exon per million reads mapped for the most frequently mutated genes in MDS MSCs of this study. This analysis showed that the most frequently mutated genes such as *TTN*, *MUC16*, or *LRP2* were not expressed in MDS MSCs. Vice versa, we correlated the mutational matrix against genes that showed significant expression in expanded MSCs. This set of genes was mostly comprised of structural proteins including *Collagen 6A3* and *4A2* (mutated in 6 and 4 cases, respectively), *Filamin B* (*FLNB*, 5 mutated cases), *Tenascin C* (*TNC*, 4 mutated cases). Yet, as predicted by MutSigCV, such mutations most likely resemble non-specific mutational hits ($p > 0.2$) and were only considered to have low to moderate impact on protein structure.

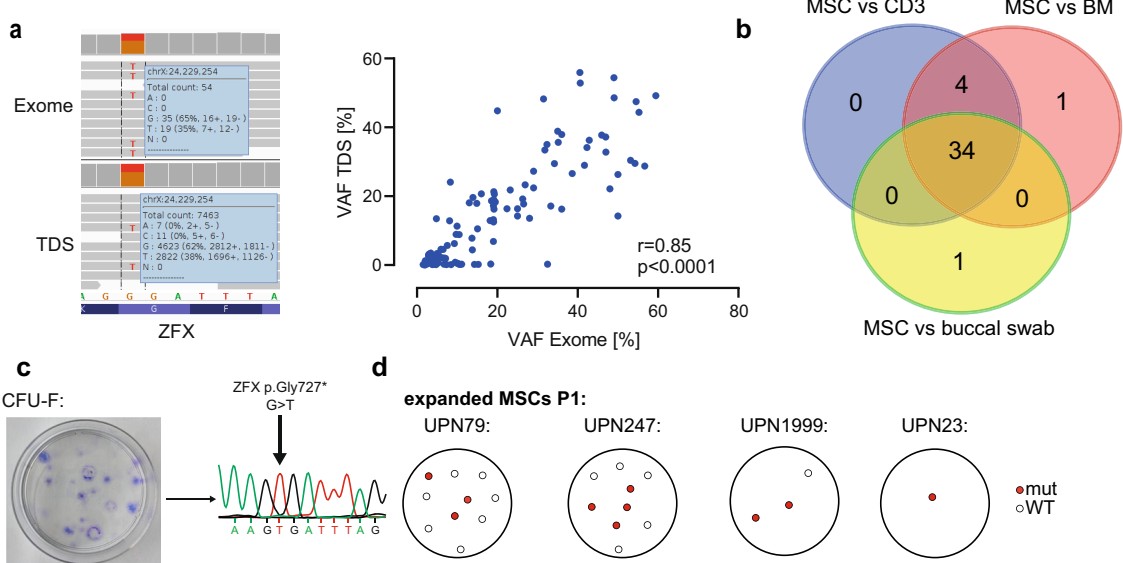

**Fig. 2 Validation of mutations. a** left: Exemplary IGV visualization of a ZFX mutation from patient UPN1999 for exome sequencing (top) and targeted deep sequencing (TDS, bottom), right: Correlation of $n = 120$ mutational calls from exome sequencing and targeted deep sequencing using PCR amplicon-based library generation and deep sequencing with minimum 300× coverage, Pearson $r = 0.85$, two-sided, $p < 0.0001$. **b** Mutations detected in MSC depending on the germline control used. For this case, MSCs, Bone marrow mononuclear cells (BM), CD3+ FACS purified cells from peripheral blood (CD3), and buccal swap underwent independent exome sequencing and independent mutational calling was carried out and filtered for high-quality mutations with VAFs >10%. The respective overlap is shown. **c** Exemplary Sanger sequencing trace from one CFU-F derived from patient UPN1999 with the ZFX mutation from (**a**). **d** Visualization of CFU-F generated from three patients and genotyped by Sanger sequencing for high confident mutations present in expanded MSCs. Colonies were color-coded if at least one mutation from several tested was present in the respective colony. Source data are provided as a Source Data file.

**The occurrence of MSC specific mutations is related to in vitro culture.** Since most of the detected acquired mutations in the MSCs were subclonal, we interrogated their dependency on culture parameters to address to which extent mutations in MSCs were possibly a result of selection during in vitro culture. We found a moderately positive correlation of the number of mutations with the duration of in vitro expansion of MDS-derived MSCs (Fig. 4a; $r = 0.25$, $p = 0.03$). Moreover, primary MDS MSCs that were cultured for an additional passage (P2) showed an increased number of mutations with VAFs >10% (Fig. 4b; median 26 (P1) vs. 90 (P2) mutations/sample, $p < 0.0001$).

To further and more directly address the question of the dynamics of mutational acquisition and stability in MSC culture, we analyzed mutational profiles during serial in vitro culture and compared chronological passages P0 and P1 of the same cultures, and calculated mutational clusters using the SciClone tool[37]. Thereby, we derived distinct clones based on independently sequenced serial samples (Fig. 4c–i). While there were some cases, in which we could detect relatively prominent and stable mutations with VAFs > 30% in both passages (Fig. 4c, d) (ZFX p.Gly727, DNAH7 p.G862C), most mutational clusters were characterized by low VAFs (<25%) and exclusiveness for either the early or the later passage (Fig. 4e–i). In addition, we performed sequential exome sequencing of two MSC cultures from the same patients from independent BM aspirations in five cases. We found that the mutations were mutually exclusive in all 5 cases, also for frequently mutated genes such as e.g., TTN or DCH2. Nevertheless, mutational signatures clustered pairwise for the individual patients in four of the five sample pairs (Fig. 4j). This observation further supported the notion that patient-specific conditions determine the mutational signatures. Collectively, these data indicate that the majority of mutations were expanded randomly and highly dynamically during the course of the in vitro expansion of the MSCs.

**MSC culture-specific mutations cannot be backtracked to primary bone marrow stroma cells.** While the serial timepoint analyses of MSCs in culture revealed that most mutations in subpopulations were expanded during in vitro culture, we still aimed to pursue the relevance of higher confidence mutations in vivo in humans. Mutations such as in ZFX, DNAH7, and others (Fig. 4c, d) could have expanded from pre-existing non-hematopoietic clones in the BM with higher VAFs and remained stable in culture. We, therefore, performed another targeted sequencing approach to validate and backtrack these mutations in primary, non-expanded stroma cells of the same patients. We sorted viable CD45− CD235a − CD31− CD271+ cells from primary BM aspirates (Fig. 5a) from the same BM aspiration and subjected them to targeted re-sequencing. This procedure confirmed previously published experiences of inherent difficulties to isolate sufficient numbers of BM stroma cells from primary BM samples[11]. From $n = 7$ cases we were able to sort a median of 7829 cells per patient (range 417–74,698). Since low cell numbers in this range are even challenging to re-sequence in a targeted amplicon-based NGS approach, we have previously verified that we could technically quantify low VAF SNPs in samples with low cell number input material[26]. In addition, we analyzed DNA from a less stringent isolation strategy comprising CD45−, CD235a−, CD271+/−, CD31+/− cells, therefore also including the CD31+ endothelial cell population (Fig. 5a) for $n = 9$ cases. We performed this targeted approach to sort these subpopulations from primary BM for a total of nine cases with 1–9 high confidence mutations each (Fig. 5b–j). For some patients, we were also able to perform this analysis from chronologically later BM specimen 1–3 years later after the initial sample acquisition (Fig. 5g, j). In all cases, high confidence mutations detectable in ex vivo MSC cultures were neither significantly detectable in the stringently primary sorted CD45− CD235a−, CD271+, CD31− stromal cell fraction nor the less stringently sorted non-hematopoietic fraction including endothelial cells (CD45− CD235a− CD271+/−, CD31+/

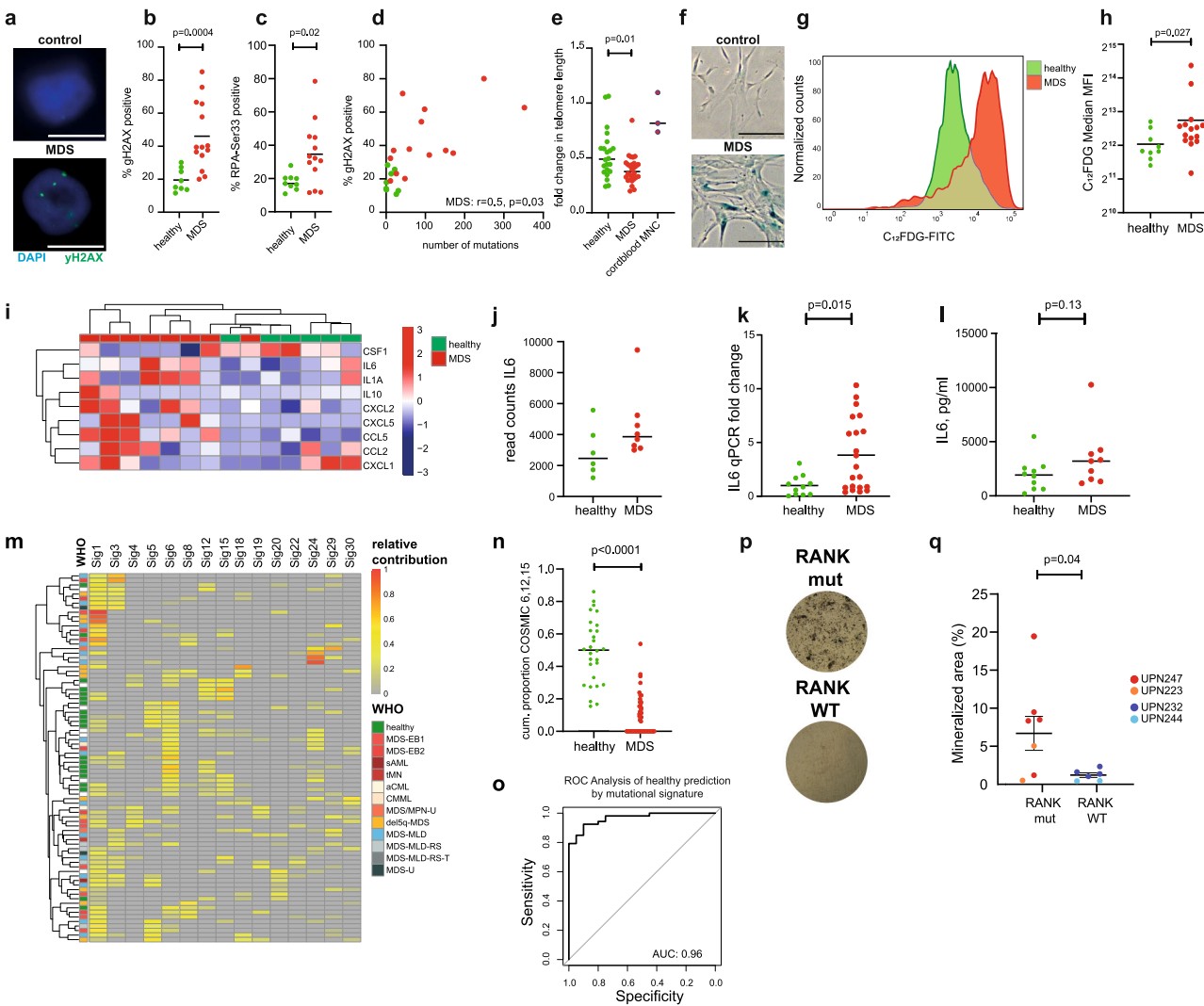

**Fig. 3 Phenotypic differences in mutated MSCs. a** Representative immunofluorescence images from MDS and control MSCs upon γH2AX staining. DAPI in blue, γH2AX antibody in green, scale bar represents 10 μm. **b, c** Quantification of γH2AX and RPA foci in at least 50 cells analyzed per sample, two-sided Mann–Whitney test. Data are presented as mean and individual values. **d** Correlation of γH2AX foci and number of mutations in exome sequencing. Pearson r, two-sided p value. **e** Quantification of the relative difference in telomere length in MDS and control MSCs using qPCR to interrogate n = 28 MDS, n = 22 healthy, and n = 3 cord blood, biologically independent mononuclear cell (MNC) samples, two-sided Mann–Whitney test. Data are presented as mean and individual values. **f, g** Representative β-galactosidase staining, and flow cytometry result of C$_{12}$FDG, a fluorogenic substrate of β-galactosidase. scale bar represents 100 μm. **h** Quantification of C$_{12}$FDG mean fluorescence intensity (MFI) for MDS and control MSCs, two-sided Mann–Whitney test. Data are presented as mean and individual values. **i** Gene expression heatmap of genes associated with the inflammatory senescence-associated secretory phenotype (SASP) derived from RNA sequencing. **j** RNA Sequencing read counts for IL6 in expanded MSCs. Data are presented as mean and individual values. **k, l** qPCR and ELISA validation of IL6 expression in a cohort or n = 32 MDS- and 19 healthy MSCs, two-sided Mann–Whitney test. Data are presented as mean and individual values. **m** COSMIC mutational signatures in MSC samples with >20 mutations for signatures and more than a total 10% occurrence in the cohort. Relative contribution is color-coded. Sig: COSMIC Signature. **n** Cumulative (cum.) proportion of COSMIC signatures 6, 12, and 15 per group, two-sided Mann–Whitney test. Data are presented as median and individual values. **o** Receiver operating curve (ROC) for classification of MDS and healthy according to COSMIC signatures 6, 12, and 15. AUC area under the curve. **p, q** Quantification of ossification as quantified by von Kossa-Staining upon a 21d osteogenic differentiation protocol of MDS-MSCs with or without RANK mutation, each dot represents an independent differentiated aliquot. two-sided Mann–Whitney test. Data are presented as mean and individual values, error bars represent SEM. Mutations were confirmed in differentiated osteoblasts by Sanger sequencing. Source data are provided as a Source Data file.

−), or in the corresponding hematopoietic fraction. Finally, for four patients we replated BM from the same bone marrow aspiration used for initiation of the exome sequenced bulk cultures. In DNA from a total of 34 CFU-F colonies, we re-sequenced a total of 24 high confidence genes including RANK and ZFX. Only in 1/34 (3%) CFU-Fs, a ZFX mutation was detectable (Table 3), thus further confirming that even the highest confidence mutations detected in ex vivo expanded MDS MSCs did not originate from clonal stroma cell populations in vivo.

## Discussion

The scientific background and rationale for this investigation were an increasing body of evidence suggesting the pathogenesis of MDS and other myeloid neoplasms such as MPN or AML not only takes place in the hematopoietic compartment but also in the BM micro-environment. This conception is supported by a multitude of different studies and approaches describing altered phenotypes of bone marrow stroma components of malignant myeloid disease such as features of increased senescence and impaired proliferation

**Table 2 RNA seq. gene-expression data from MDS MSCs.**

| Ensemble ID | Gene symbol | Mutated cases | Mean FPKM in in vitro MSC |
|---|---|---|---|
| ENSG00000198034 | RPS4X | 4 | 471 |
| ENSG00000163359 | COL6A3 | 6 | 357 |
| ENSG00000134871 | COL4A2 | 4 | 316 |
| ENSG00000041982 | TNC | 4 | 153 |
| ENSG00000136068 | FLNB | 5 | 54 |
| ENSG00000005889 | ZFX | 8 | 23 |
| ENSG00000115317 | HTRA2 | 4 | 17 |
| ENSG00000130702 | LAMA5 | 8 | 1.11 |
| ENSG00000154358 | OBSCN | 8 | 1.06 |
| ENSG00000155657 | TTN | 17 | 0.09 |
| ENSG00000141655 | RANK | 5 | 0.08 |
| ENSG00000081479 | LRP2 | 10 | 0.03 |
| ENSG00000181143 | MUC16 | 10 | 0 |
| ENSG00000173976 | RAX2 | 4 | 0 |

FPKM indicates fragments per kilobase of exon per million reads mapped.

capacity[11,30,38,39], increased inflammatory phenotypes[11,40,41], globally transcriptionally[10,11,31], and epigenetically aberrant profiles[14,30,42] and hematopoietic support[12,30,43,44] (reviewed in refs. [4,5,45]). Moreover, several genetic mouse models have repeatedly shown proof of principle data that isolated genetic manipulation of BM niche components could induce MDS-like or myeloproliferative diseases and influence propagation of malignant hematopoietic clones[15–18,46]. Therefore, we here aimed to clarify a long-standing question, as to whether primary MSCs of MDS patients carry somatically acquired mutations with clonal relevance in vivo.

Several lines of work have addressed this hypothesis before and revealed chromosomal and mutational events in MSCs derived from MDS or AML patients[24,25,47]. However, it has never been entirely ruled out that such molecular lesions were not secondary to the highly expansive in vitro cultures, which are necessary to obtain these cells.

The ideal experiment to address the question of acquired mutations in BM stroma of MDS patients would therefore be explorative sequencing performed in sorted primary non-hematopoietic cells from primary bone marrow samples. However, to date, there is still uncertainty on the immunophenotypic profiles the target cells for this experiment should have. Human BM-derived mesenchymal stem cells have previously been defined as CD45−, CD235a−, CD31−, and CD271+, CD146+, CD105+[18]. However, from the few studies that have performed molecular analyses on such highly purified and scarce cells it has become clear, that so far, it has not been successful to reproducibly isolate enough of these cells to perform robust explorative whole-genome or exome sequencing on a representative cohort. Therefore, also in our study, we have made the compromise to begin analysis on in vitro expanded MSCs of MDS patients. To this end, we have analyzed a representative cohort of $n = 98$ MDS and myeloid neoplasia derived BM MSCs as well as $n = 28$ healthy controls and used their own paired hematopoietic cells as germline controls. By standard bioinformatic evaluation, accounting for LOH in the hematopoietic germline fraction, we detected a large number of acquired mutations in genomes of ex vivo expanded MDS MSCs and even enrichment of recurrent events with frequencies of up to 19%. Interestingly, MDS MSCs clearly presented with more mutations and a higher mutational burden than the healthy control group suggesting that MDS or myeloid neoplasia-derived MSCs may be functionally altered in comparison to healthy MSCs. We performed several lines of characterization experiments with the MSCs from our study demonstrating that MDS-derived MSCs had indicators for higher replicative stress, increased senescence, and increased levels of inflammatory IL-6 as compared to healthy MSCs. Moreover, MDS MSCs had significantly different COSMIC mutational acquisition signatures[35], which possibly resulted from mutational processes specific to in vitro cultivation[48]. Together with previously published data of increased levels of reactive oxygen species (ROS) in the BM niche of myeloid neoplasms[49–53], these results corroborate the above-described notion that MSCs in myeloid neoplasia are molecularly and functionally altered. The observed replicative stress, activation by inflammation, and ROS may induce higher cell division rates and increased baseline genomic instability of MSCs derived from myeloid diseases and therefore explain higher disease-associated mutagenesis.

Most of the detected mutational events in MSCs had low-level VAFs and were mostly accumulated in sites with higher mutability during the in vitro expansion such as, e.g., TTN[27], which is known to be frequently mutated in explorative WES studies and most likely does not represent a clonal driver mutation. Therefore, we hypothesized that these events were largely secondary due to expansion in culture rather than true driver lesions originating from clonal mutations in the non-hematopoietic BM compartment. To follow this hypothesis we performed additional experiments that asked the question whether patient individual profiles detected in standard MSCs cultures (P1) were also measurable in the earliest possible in vitro culture passages (P0) and whether they remained stable in measurements of serial culture passages. Both approaches revealed that most mutational events occurred in minor clone sizes and were largely not even stably passed on from P0 to P1, precluding them from the potential discovery of real driver mutations. Nevertheless, this was not necessarily applicable to all mutations. For instance, ZFX stood out as a possible true mutational driver event due to its recurrence ($n = 8$ cases), its higher VAF of up to ~35%, and its stable detectability during serial cultures. The same was true for other high confidence candidate mutations such as DCHS1 or RANK[54,55], of which the latter also had a functional impact on in vitro osteogenic capacity in our hands, which was in line with previous work[56,57]. Due to the putatively higher relevance of these mutations, we attempted to backtrack them in the primary BM of the patients, in which they were detected. These high confidence candidate mutations were neither detectable via TDS in the unselected whole bone marrow samples, nor in the primary sorted and non-expanded CD45-, CD235a-, CD31- and CD271 + cells as well as the less stringently sorted CD45−, CD235a−, CD271+/−, CD31+/− fractions. We have previously confirmed the technical validity of our applied amplicon-based deep re-sequencing approach on low DNA amounts from low cell number samples by serial dilution spike-in experiments[26]. As an alternative method, we also re-cultured the original MSC samples by CFU-F assays and performed genotyping of the candidate mutations in single colony sequencing. Thereby, the top recurrent candidate mutations could also not be detected. From these results, we finally concluded that there were no relevant clonal mutations in the BM stroma fractions of these MDS patients and that even the high confidence mutations detected in serial MSC culture samples rather reflect outgrowth of particularly fit clones originating probably of single cells selected by the culture conditions. This is also in line with previous approaches using lentiviral barcoding, which have shown that culture expansion of MSCs is associated with massive clonal selection and loss of clonal complexity[58]. Since previous mechanistic experiments in murine models had shown proof of concept that ectopic molecular alterations of the BM niche could induce MDS like phenotypes, we also interrogated, whether we could detect stromal alterations known to induce MDS in murine models such as PTPN11[17], β-catenin[16], DICER[15], Sbds[15], or RARγ[59] in MDS MSCs but found no relevant mutations in these genes in our data.

While we ruled out the evidence for acquired clonal mutations in the stroma compartment of MDS patients, the possibility for other molecular differences such as increased inflammatory status, epigenetic, transcriptional, senescence, and disturbed niche

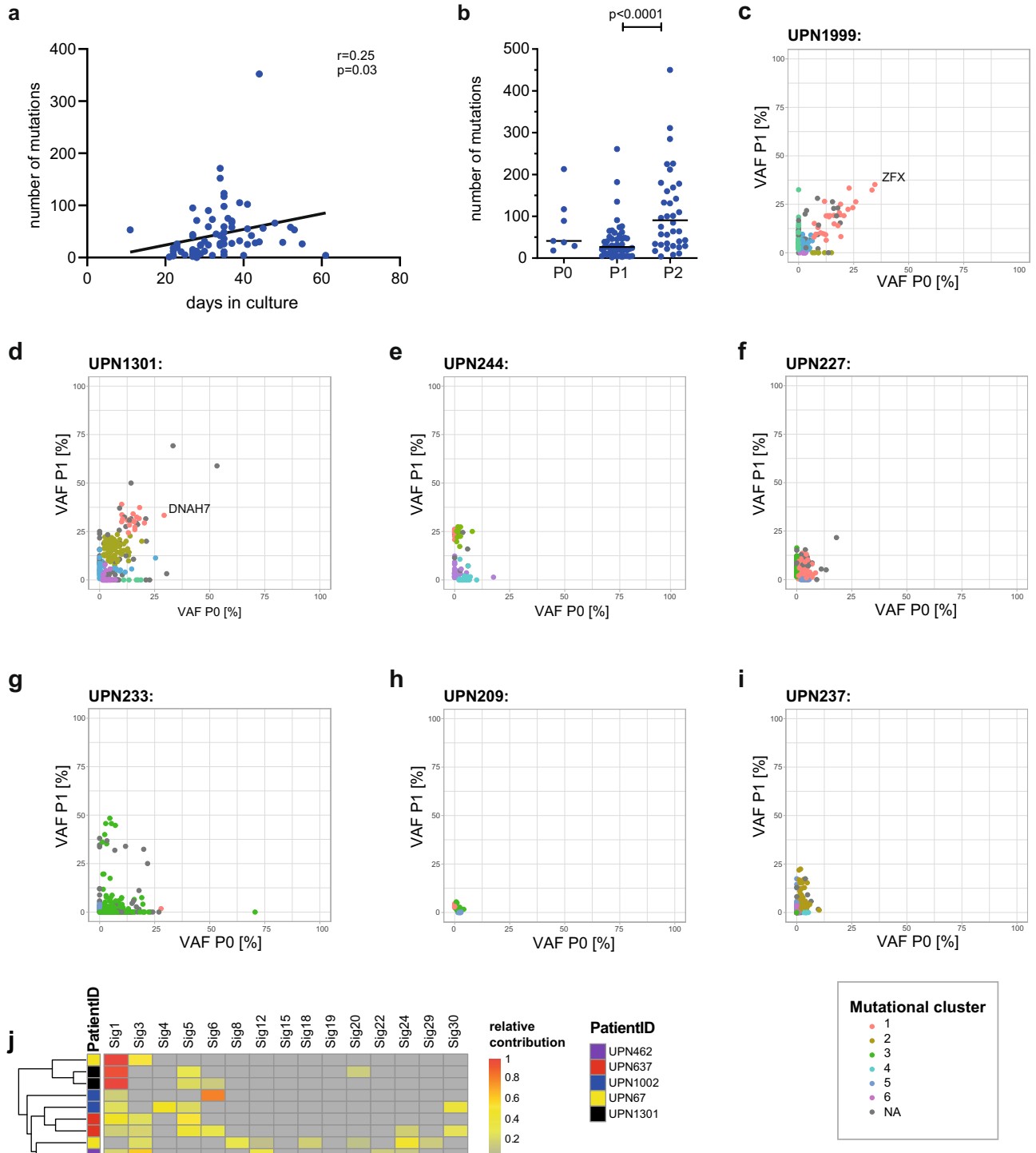

**Fig. 4 Time series analysis of mutations during culture. a** Spearman correlation of the total number of mutations associated with the time from seeding to harvest. **b** Total number of mutations in dependency of the culture passage, $n = 7$ (P0), $n = 48$ (P1), and $n = 38$ (P2) biologically independent samples, two-sided Mann–Whitey test, P = passage. Data are presented as median and individual values. **c–i** VAFs of individual mutations obtained from two different passages of the same culture derived from exome sequencing of $n = 7$ MDS cases. Color-coded mutational clusters were calculated with the sciclone tool[37]. **j** Extracted mutational COSMIC signatures for $n = 5$ patient sample pairs upon exome sequencing from 2 independent bone marrow aspirations. Patient IDs are color-coded. Source data are provided as a Source Data file.

interaction of MDS derived MSCs, of course, remain[6,7,9–14,30,60]. Furthermore, from our data, we cannot exclude the possibility that on a confined local level of the bone marrow there may be clusters of clonally mutated stroma cells.

Nevertheless, collectively this comprehensive analysis leaves little doubt that if acquired mutations in the stroma of MDS patients play a role in MDS disease initiation at all, then at such a low clonal and possibly locally confined level, that they are not detectable with

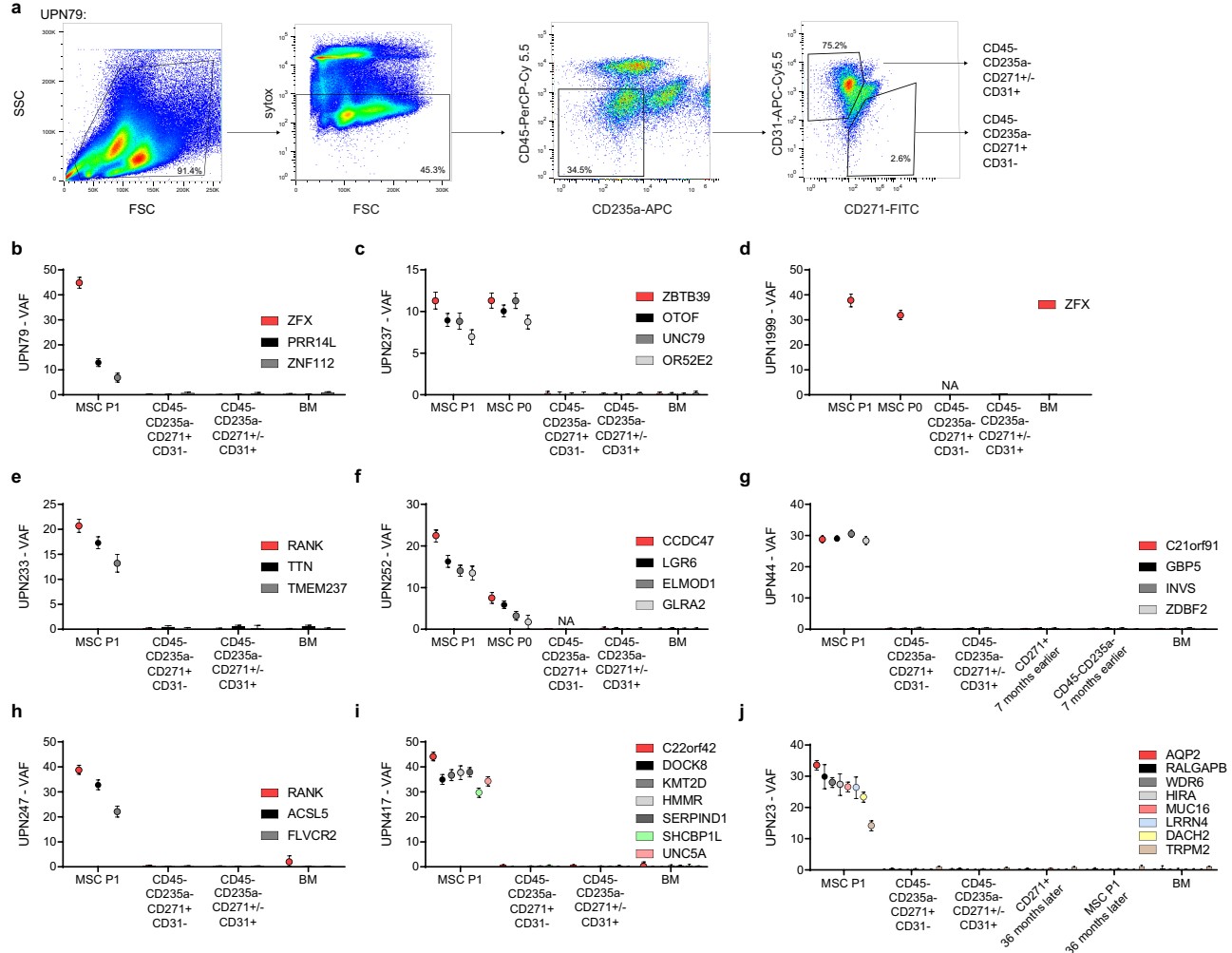

**Fig. 5 Backtracking of mutations in primary sorted MSCs. a** Exemplary FACS gating scheme (see also Supplemental Fig. 2) for the sort of primary viable CD45− CD235a− CD31− CD271+ MSCs and a less stringent non-hematopoietic CD45−, CD235a−, CD271+/−, CD31+/− fraction. **b**–**j** Comparison of selected mutations with significant VAFs in expanded MSCs versus non-expanded primary sorted MSCs with custom PCR amplicon sequencing with a minimum coverage of 400× for nine cases. Data are presented as VAF and error bars represent ±95% confidence interval derived from sequencing coverage determined using equal or given proportions-test with continuity correction; NA, not available due to less than 125 sorted cells available. Source data are provided as a Source Data file.

**Table 3 Genotyping of CFU-f formed by seeding of BM MNCs from culture initiating BM aspirations.**

| Patient | Mutations tested | CFU-F | |
|---|---|---|---|
| | | WT | mut |
| UPN79 | ZFX, PRR14L, ZNF112 | 9 | 1 (ZFX only) |
| UPN417 | C22orf42, DOCK9, HMMR, KMT2D, MMP27, SERPIND1, SHCBP1L, UNC5A, ZNF853 | 6 | 0 |
| UPN247 | RANK, ACSL5, FLVCR5 | 10 | 0 |
| UPN23 | AQP2, RALGAPB, WDR2, TRPM2, MUC16, SH3BP2, LRRN4, HIRA, DACH2 | 8 | 0 |

currently feasible sample acquisition and methodology. In our current study, we discovered no evidence for acquired mutations as disease initiators for MDS.

## Methods
**Patient and healthy donor samples.** The study cohort consisted of $n = 98$ MDS and MDS associated myeloid neoplasia cases, who were treated at the Department

of Hematology and Oncology of the Medical Faculty Mannheim, Heidelberg University, Germany (median age 73 years, range: 44–86). As healthy controls, a cohort of $n = 28$ healthy BM samples was acquired from patients undergoing hip replacement surgery (median age 75 years, range: 36–84). These healthy donor samples all had normal blood counts, absence of active or prior malignancy, and other confounding co-morbidity. All experiments were performed after written informed consent in accordance with the Declaration of Helsinki and approved by the Medical Ethics Committee of the Medical Faculty Mannheim. Detailed clinical characteristics of patients and healthy donors are provided in Table 1.

**MSC cultivation.** MSCs were expanded adherently on plastic dishes by seeding 100 μm-filtered full BM fragments and additionally seeding $5 \times 10e6$ mononuclear cells in StemMACS MSC Expansion Medium XF (Miltenyi Biotec) in T25 flasks (P0). After 2 days, the medium was changed and non-adhesive cells were removed. Cells were then further expanded with weekly medium changes. At 80% confluency, this initial culture was trypsinized and split into 2–4 T75 flasks, corresponding to P1 with yields of approximately 60,000 cells per flask. Of note, the in vitro culture was carried out for the shortest possible period of time to obtain sufficient cell numbers for bulk DNA isolation. Cells were harvested before senescence or confluence. The median time of in vitro expansion before DNA preparation was 34 days, (95% confidence interval (CI): 22–50 d).

**Whole-exome sequencing.** A subset of $n = 45$ MSC exome sequence data sets of this cohort was previously used as germline control for mutational analyses of BM in a prior study[26]. The additional $n = 43$ cases were sequenced de novo for the current study. To interrogate somatically acquired mutations in MSCs we reversed

the bioinformatic settings and defined the hematopoietic cells as germline controls, which is described in detail in the supplemental methods section. Median coverage of exome sequencing was 81× (range 19–140×) for the MDS MSCs and 164× (range: 118×–215×) for the healthy cohort.

Mutational significance was calculated with MutSigCV (v1.4)[27] from maf files after vcf2maf (https://github.com/mskcc/vcf2maf) conversion from the mskcc repository. We used publicly available context files for coverage, covariates, and mutational dictionaries. Mutational signatures were extracted using the vcfs generated previously according to Maura et al.[61] using the R (v3.6.3) MutationalPattern (v1.10) package[62]. For extraction of COSMIC signatures whichSignatures from the deconstructSigs[63] (v1.9) were applied to vcfs with context normalization and default trinucleotide counts.

**RNA sequencing.** For RNA sequencing analyses, polyA libraries from $n = 5$ MDS samples and $n = 3$ healthy sequenced in a previous study[10] were combined with de novo sequenced samples from an additional $n = 3$ MDS patients and $n = 3$ healthy donors in the current study. All samples were prepared equally and finally added up to a combined cohort of $n = 8$ MDS samples and $n = 6$ healthy samples. In brief, 500 ng RNA was subjected to the Illumina Stranded TruSeq RNA protocol and sequenced to a median of 94 million reads per sample. The bioinformatic analysis consisted of mapping hisat2 v2.04 to hg38 and cufflinks—cuffdiff v2.2.1 for transcript assembly and differentially expressed gene analysis according to the vignette with default settings.

**Sorting of primary MSCs and deep sequencing validation of candidate mutations.** Validation of high confidence candidate mutations was carried out in the original samples as well as in primary, non-expanded MSCs, which were FACS sorted from viably frozen BM specimen of the corresponding patient samples, in whose MSC cultures the mutations were detected. Viable cells were stained and sorted for the parameters Sytox®−, CD45−, CD235a−, CD31−, CD271+ (CD45-HI30, BD Bioscience, PerCP-Cy 5.5, Cat No: 564106, 1:100; CD235a-GA-R2, BD Bioscience, APC, Cat No 551336, 1:100; CD31-WM59, Biolegend, APC.C7 Cat No 56365, 1:1000; CD271-ME20.4, Biolegend, FITC Cat No 345104, 1:20; Sytox®, Thermo Fisher) on a BD FACS Melody sorting device. In addition, a second FACS strategy (Sytox−, CD45−, CD235a−, CD31+/−) was employed to enrich non-hematopoietic cells. Cells were directly sorted into Qiagen ALT lysis buffer and the whole genome amplified with the Qiagen repliG Kit in the majority of cases. For validation with targeted deep sequencing, single PCRs surrounding mutational sites (Primers in Supplemental Data 1) were carried out with subsequent library generation using the Nextera XT kit (Illumina). Samples were pooled and sequenced with MiSeq v3 chemistry at a mean coverage of 8699x. Quantification was then derived from bamfiles after bwa mem (v0.7) and picard MarkDuplicatevs (2.20). For further detailed methods, please see the Supplemental methods section.

**Reporting summary.** Further information on research design is available in the Nature Research Reporting Summary linked to this article.

## Data availability

The RNA Sequencing, Exome-Sequencing and targeted resequencing data generated in this study have been deposited in the EGA archive under accession codes EGAD00001006968 (exome sequencing, RNA sequencing, and targeted resequencing of this study) and EGAS00001000716 (RNA sequencing reported by[10]). These data are available under restricted access for scientific research-only use. Access can be obtained through Daniel.Nowak@medma.uni-heidelberg.de. Responses can be expected within 72 h. Access can also be requested via EGA. Source data are provided with this paper.

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

## Acknowledgements

This work was supported by funds of the Deutsche Forschungsgemeinschaft (DFG) (No. 817/5-2, FOR2033, NICHEM), the German cancer aid foundation (Deutsche Krebshilfe, 70113953), the Gutermuth Foundation, the H.W. & J. Hector foundation (Weinheim) (Project M83), the Dr. Rolf M. Schwiete Foundation (Mannheim), the Wilhelm Sander Foundation (2020.089.1) D.N. is an endowed Professor of the German José-Carreras-Foundation (DJCLSH03/01). We thank the Institute of Transfusion Medicine and Immunology, Medical Faculty Mannheim, Heidelberg University; German Red Cross Blood Service Baden-Württemberg—Hessen for providing access to the microscope.

## Author contributions

J.-C.J., D.N. and W.-K.H. designed the study, analyzed the data, and wrote the paper; J.-C.J., M.M. and D.N. conducted the experimental design, executed bioinformatic analyses, and most of the experiments; D.N. and W.-K.H. supervised the whole study and provided the research infrastructure; V.N., V.R., J.O., I.P., A.S., J.F., E.A., N.S., V.R., and X.Q. performed molecular or FACS analyses and functional experiments. J.-C.J., F.N., A.J., A.D., M.J., G.M. and D.N. provided the patient material and clinical data.

## Funding

## Competing interests

The authors declare no competing interests.
