## [Peer review file · Nature Communications]

REVIEWER COMMENTS

Reviewer #1 (Remarks to the Author): Expert in MDS stem cells and genomics

In their manuscript "Bone marrow derived stromal cells from myelodysplastic syndromes are altered but not clonally mutated in vivo" Jann et al. systematically dissect the mutational status of MDS-patient derived stromal cells compared to their in vivo counterparts.

Studies in the last years have clearly highlighted a contribution of the stroma to the pathogenesis of MDS in terms of a niche-facilitated leukemogenesis. However, so far the mutational status of stromal cells remained speculative, so the elegant data by Jann et al. provide an important insight. It is interesting, and surprising, that MDS-MSCs seem to be more susceptible to acquire mutations in vitro but do not really show mutations in the in vivo situation.

The absence of mutations in stromal cells in MDS is the central and most important finding of this study. The absence of mutations in stromal cells does not exclude molecular alterations by other mechanisms like inflammation as the authors also discuss on page 14.

The study is well-controlled and well written.

However, the impact of mutational alterations in cultured MDS-MSCs remains elusive. It is quite striking that MDS-MSCs seem to be more susceptible to acquire mutations when in culture. The title also states that MSCs in MDS patients are altered. This part was not really clear to me. As in the current form of the manuscript, I would suggest to include more functional analyses to understand these findings.

1. Genotoxic stress in the HSC-stromal cell interaction was described to play a central role in the pathogenesis of MDS. Could this be an explanation for a higher susceptibility to acquire mutations?
2. Can the authors compare ROS levels, DNA damage etc. in cultured MSCs from MDS patients compared to healthy donors. Is there a correlation to mutation?
3. Do mutated MSCs from MDS patients have a more "inflammatory" phenotype?
4. Do MSCs differ in surface markers, functional properties as differentiation.
5. How does hematopoiesis support change in mutated MSCs? Are there functional differences. Do the authors think that using MSCs isolated from MDS patients for functional studies in general is a problem? Are these cells still a good model system.
6. The authors state in the abstract that the mutations identified in the cultured stromal cells are "artefacts". Why are they then more likely to occur in MDS patients? Is there a senescence phenotype? What is the telomere length?
7. The title suggests that MSCs in MDS are altered in vivo but this is not really matter of subject here. In its current form I would change the title and make a stronger statement that MSCs are not clonally mutated in MDS in vivo.
8. Could MSCs be sorted as CFU-F and mutational status be tested? This would allow to define more "clones" in a less artificial environment than pure culture.

Reviewer #2 (Remarks to the Author): Expert in bone marrow microenvironment, stem cells and MSCs

The manuscript by Jann J-C et al is an interesting study on the possible meaning of mutations found in mesenchymal stromal cells (MSCs) from MDS patients, compared with healthy controls.

The authors perform whole exome sequencing of expanded MSCs and targeted mutation analysis of primary samples. They confirm the presence of mutations in cultured MSCs previously reported in other studies. They discover multiple recurrent mutations but are unable to find most of these in the primary samples, although the number of cells available might be limiting. The authors conclude that

the discovered mutations are culture artefacts and that there is no evidence for clonal mutations in the BM stroma of MDS patients. However, there are some technical limitations in the study, a wide age variability does not appear to be considered, the potential functional implications of recurrent mutations in MSCs or their progeny is not tested and the fact that the mutations are recurrent, some of them affect important pathways in MSC and bone-forming cells (such as RANK), should be considered as a note of caution when interpreting the results.

Specific comments:

1. The number of mutations is known to increase with age. The cohorts of MDS and healthy individual have 6-year difference in median age and a wide variability in age (36-86) which does not seem to be accounted for. Age should be included as an important factor in the equation.

2. There is an inherent technical limitation in the ability to detect mutations dependent upon the number of cells (could this be measured?) and this might explain differences between primary cells and expanded cells needs to be taken into account both as a limitation of the study and in the interpretation of the results. In Figure 6E, although independent samples of the same patients show that 4 out of 5 patients cluster together in the two samples, there is significant variability in the mutations detected for the same patient...does this variability also affects the recurrent mutations found?

3. The authors use the corresponding hematopoietic fraction as a germline control for the MSC exome sequencing data. However this reduces coverage since the authors have to exclude by hematopoietic affected genomic regions from analysis in the respective samples on an individual basis. How large are the genomic areas excluded and are there no mutations at all present in MSCs in these areas? Is it possible that excluding different genomic areas in different patients (according to their mutational status) contributes to the observed heterogeneity? A more traditional germline control (not bearing known oncogenic drivers) might overcome these potential issues or limitations.

4. The correlation between the number of mutation and the duration of in vitro expansion (Figure 3C) does not seem as clear as claimed ($r=0.42$)

5. P. 13. Related to "We therefore hypothesize that while previous mechanistic experiments in murine models have shown proof of concept that ectopic molecular alterations of the BM niche can induce MDS like phenotypes, this is not a relevant pathomechanism in adult human MDS." This seems an overstatement... As the authors mention, although no clonal mutations were detectable in this study, other molecular alterations/signatures may in the MDS BM stroma may be pathogenic in MDS... Otherwise why would MSCs support and increase MDS HSC engraftment in previous studies? How to explain a higher mutational susceptibility of MDS MSCs? Additionally, there are germline mutations e.g. SBDS, PTPN11 associated with MDS/JMML development and their presence in the stroma has been shown to contribute to disease pathogenesis.

6. P. 13. Related to "Therefore, we hypothesized that these events were largely secondary culture artefacts rather than true driver lesions originating from clonal mutations in the non-hematopoietic BM compartment" and "Discovery of valid mutations". In the absence of functional studies assessing the impact of these recurrent mutations it is hard to judge whether they are functionally relevant or not. For instance, functional mutations that affect RANK might have important consequences in how these cells proliferate or differentiate in osteoblastic cells, and thereby influence their support of malignant hematopoietic cells. Although the authors claim that "even the high confidence mutations detected in serial MSC culture samples rather reflect outgrowth of particularly fit clones originating probably of single cells selected by the culture condition", the fact that these mutations are recurrent or selected in culture suggests a biological role that might secondarily affect the function of the malignant hematopoietic cells.

7. The different COSMIC signatures and their potential meaning need to be explained.

8. What are the characteristics of UPN1999 that make explain this case standing out (age, mutational status, therapy etc)?

Reviewer #3 (Remarks to the Author): Expert in MDS genomics

In their manuscript, Jann, et al. investigate the possibility of clonal expansion in bone marrow environment in MDS patients, a long-standing issue in the field of MDS on the basis of several observations in human and in mouse. The authors analyzed in vitro-cultured BM-derived MSCs from both normal individuals (n=20) and patients with MDS (n=98) using whole exome sequencing to detect somatic mutations using BM hematopoietic cells as control. Serially obtained samples over 1-3 years were analyzed in 9 cases. The presence of detected mutations in in vitro cultured cells were tested in primary samples. The authors detected somatic mutations in samples from MDS patients and normal individuals with unknown frequencies. Several genes, including ZFX, RNAK, TNFRSF11A, HTRA2, and PBM44, were significantly recurrently mutated. All mutations are found in a small subset of cultured cells with significantly larger VAFs in MDS-derived samples. However, most of these mutations were not stably detected during in vitro passages or detected in primary samples. On the basis of these observations, the authors concluded that these mutations are related to in vitro cell culture but does not represent clones positively selected in vivo. Although the conclusion is negative one, these observations are of interest worthwhile reporting. Meanwhile, several issues need to be addressed before it is considered for publications.

Major issues:

- 1) It should be clearly stated how many MDS and normal MSC samples had somatic mutations? Authors just reported distributions of VAFs.
- 2) Mutation calls in whole exome sequencing need to be validated at least for a subset of mutations for varying VAFs. This is critical.
- 3) As the authors mentioned, some control samples (i.e., MDS cells) were expected to have many copy number abnormalities, particularly LOH and should have complicated the detection of MSC-specific alterations. However, no details of the mutation calling methods were described. Please clarify how to detect MSC-specific mutations.
- 4) Were there any mutational signatures that suggested an influence from in vitro culture?
- 5) Fig. 6A: it is problematic to decompose a small number of mutations in many cases, which may not represent real mutational processes. Thus, this may be misleading. So cases having small numbers of mutations, particularly <20 should be deleted.
- 6) Although recurrent mutations identified in in vitro-cultured cells were not confirmed in primary MSCs, it might be informative for authors to clarify the limitation in sensitivity in their study. In other words, to what extent can the authors exclude the possibility of the presence of such mutations.

Minor issues:

- 7) Meaning of sentences is unclear here and there. Followings are just among many examples:
 - P9: Vice versa, we correlated ...
 - P9: These mutations most likely resembled...
 - P9: ...calculated distinct clones...: clones cannot be calculated.
 - P9: the last sentence.
 - Many clumsy wordings and expression.
- 8) The use of 'subclonal' is not appropriate, because the authors are using this to indicate that mutations are found in only in a subset of cells and there existed no 'major clones'. Please rephrase this wording.
- 9) English needs to be checked by a native speaker.

Rebuttal letter answered point by point

REVIEWER COMMENTS

Reviewer #1 (Remarks to the Author): Expert in MDS stem cells and genomics

In their manuscript “Bone marrow derived stromal cells from myelodysplastic syndromes are altered but not clonally mutated in vivo” Jann et al. systematically dissect the mutational status of MDS-patient derived stromal cells compared to their in vivo counterparts.

Studies in the last years have clearly highlighted a contribution of the stroma to the pathogenesis of MDS in terms of a niche-facilitated leukemogenesis. However, so far the mutational status of stromal cells remained speculative, so the elegant data by Jann et al. provide an important insight. It is interesting, and surprising, that MDS-MSC seem to be more susceptible to acquire mutations in vitro but do not really show mutations in the in vivo situation.

The absence of mutations in stromal cells in MDS is the central and most important finding of this study. The absence of mutations in stromal cells does not exclude molecular alterations by other mechanisms like inflammation as the authors also discuss on page 14.

The study is well-controlled and well written.

However, the impact of mutational alterations in cultured MDS-MSCs remains elusive. It is quite striking that MDS-MSC seem to be more susceptible to acquire mutations when in culture. The title also states that MSC in MDS patients are altered. This part was not really clear to me. As in the current form of the manuscript, I would suggest to include more functional analyses to understand these findings.

Answer: We thank the reviewer for his critical and justified judgment. We agree that the manuscript could be improved by embedding the results of our mutational data in some functional data generated in the underlying MSCs. Therefore, we have followed almost all of the helpful suggestions of the reviewer outlined and answered point by point below:

1. Genotoxic stress in the HSC-stromal cell interaction was described to play a central role in the pathogenesis of MDS. Could this be an explanation for a higher susceptibility to acquire mutations?

Answer: To address this suggestion we have performed additional experiments in form of gammaH2X and RPA foci staining in MSCs of MDS patients and healthy donors from our cohort as well as telomere-lengths analyses. Both analyses showed that MDS MSCs displayed significantly increased genomic instability as evidenced by these two assays. Results have been incorporated into new **Figure 3**.

2. Can the authors compare ROS levels, DNA damage etc. in cultured MSC from MDS patients compared to healthy donors. Is there a correlation to mutation?

Answer: We thank the reviewer for also pointing to ROS levels playing a possible pathomechanistic role in the MDS bone marrow environment. However, we opted not to address this issue by additional experiments in our revision because these would have been redundant to a multitude of previous publications demonstrating deregulation of ROS levels in the bone marrow of MDS and myeloid neoplasia: Picou F et al. Blood Advances 2019, Huang L et al. Stem Cells International 2020, Zheng Q et al. Cell Death & Disease 2018, Picou 2019, Sarhan D et al. JCI 2020, Ludin A et al. Antioxidants & Redox signaling 2014 and others. The overall consensus in these papers is that MDS

bone marrow microenvironments display higher levels of ROS – frequently also on the background of iron overload. Increased ROS levels have in turn been shown to correlate with altered osteogenic vs adipogenic differentiation and with DNA damage. We have added these citations to the discussion to make aware of the aspect of increased ROS levels in the bone marrow of myeloid neoplasia.

3. Do mutated MSCs from MDS patients have a more “inflammatory” phenotype?

Answer: We agree that inflammation is becoming increasingly relevant in MDS bone marrow biology. To address this question we have analyzed RNA Sequencing data of our MSCs and identified increased gene expression profiles of senescence associated secretory phenotype (SASP) (Faget DV, Ren Q et al. Nat Rev Cancer 2019) (**novel Figure 3I**). To validate increased inflammatory gene expression in a greater cohort of the MSCs interrogated in our study, we asserted increased IL6 levels as a key marker of increased inflammatory activation in MDS-MSCs (**novel Figure 3J-L**).

4. Do MSC differ in surface markers, functional properties as differentiation.

Answer: To address this question we have extended our FACS characterization panel and generated new comparative data from n=5 MDS derived MSC cultures versus n=5 healthy MSC cultures from samples in our study (**new supplemental figure 1**). Overall, the surface marker expression of MDS MSCs were comparable to those of the healthy MSCs with a slightly increased CD105 expression in MDS. This data is in line with previous publications interrogating the immunophenotype of MDS MSCs (Corradi G et al, 2018 Stem Cell Res Ther. and Boada M et al, 2021, Hematol Transfus Cell Ther.).

With regard to functional properties, we have specifically addressed this for the RANK mutation, because RANK is a known key player in osteoblastogenesis. In comparative functional analyses of n=2 RANK mutated MDS MSCs cultures versus n=2 non-RANK mutated MDS MSC cultures we observed a higher osteogenic propensity of RANK mutant MSCs as compared to RANK wildtype MDS MSCs (**novel Figure 3P, Q**). Differential differentiation capacity and altered support for hematopoiesis of MDS derived MSCs is in line with previous publications: Chen X et al, 2018, Bone Res.; Weickert MT et al 2021, Sci Rep; Azadniv M et al 2020, Leukemia and Geyh S et al 2013, Leukemia.

5. How does hematopoiesis support change in mutated MSCs? Are there functional differences. Do the authors think that using MSC isolated from MDS patients for functional studies in general is a problem? Are these cells still a good model system.

Answer: We thank the reviewer for this important and interesting question. Functional differences of MDS derived MSCs for the support of hematopoiesis have been demonstrated previously in several studies: Medyouf H et al. 2014, Cell Stem Cell; Geyh S et al 2016, Leukemia; Azadniv M et al 2020, Leukemia and others.

The central result of our study is that MSCs (both MDS and healthy) display outgrowth of somatic mutations and therewith gradual loss of clonal diversity during in vitro culture. The data in **updated Figure 4** demonstrates that this is extremely heterogeneous, dependent on time in culture and largely takes place on a low clonal level at least in early passages. Therefore, the question of “How does hematopoiesis support change in mutated MSCs” is difficult to address experimentally with primary cells because it will almost be impossible to prospectively generate standardized MSC cell cultures. According to our data, mutations and VAFs in any culture of MDS MSCs would not be predictable and they would be underlying highly dynamic changes of profiles and VAFs, which would

have to be controlled throughout the course of e.g. a co-culture experiment. To this end it could also not be ruled out that exposure of MSCs to HSCs would not promote selection of different MSC clones than in non-HSC-exposed MSC cultures. We believe that experiments of this scope justify a separate study and go beyond revisions of the current manuscript. Therefore, taking into account previously published data and the circumstance that an experiment to answer this question is technically almost impossible, we have refrained from performing additional experiments on this issue.

With regard to the questions “Do the authors think that using MSC isolated from MDS patients for functional studies in general is a problem? Are these cells still a good model system.” - We would like to answer as follows: Since there are currently no feasible better alternatives, we think that working with primary MDS derived MSCs is still an applicable model when taking into account that a higher rate of experimental heterogeneity generally has to be expected when working with primary cells. However, with regard to our current mutational data, it seems to be advisable to work with the earliest possible passages.

6. The authors state in the abstract that the mutations identified in the cultured stromal cells are “artefacts”. Why are they then more likely to occur in MDS patients? Is there a senescence phenotype? What is the telomere length?

Answer: We agree that the use of the term “artefacts” was not ideally suitable to describe our results and have therefore removed it throughout the manuscript and replaced it with more precise description of low level clonal outgrowths in MSC cultures. To address the other questions: “Why are they then more likely to occur in MDS patients? Is there a senescence phenotype? What is the telomere length?” - we have conducted a series of additional experiments partially also already outlined in our answers to points 1-5 of this reviewer to characterize our MSCs functionally. This led to creation of a **completely new Figure 3**. There, we have also analyzed senescence (**novel Figure 3F-H**) and telomere length (**Novel Figure 3E**) analyses of our MSCs.

7. The title suggests that MSCs in MDS are altered in vivo but this is not really matter of subject here. In its current form I would change the title and make a stronger statement that MSCs are not clonally mutated in MDS in vivo.

Answer: We thank the reviewer for the critical judgment of the title. We agree that in the original version, this title was not fully supported by experimental data but rather in reference to previous literature. However, now that we have comprehensively taken up your suggestions to generate additional experimental functional and descriptive data of the MSCs used specifically in this study (see also below), we would actually suggest to keep the title of the original version.

8. Could MSC be sorted as CFU-F and mutational status be tested? This would allow to define more “clones” in a less artificial environment than pure culture.

Answer: We thank the reviewer for this suggestion to further improve the fidelity of our data. We have accommodated this suggestion in two ways:

1. We have performed single CFU-f colony re-sequencing from expanded cultures, which were exome sequenced in our study for n=4 cases with n=23 colonies. This experiment confirmed the VAFs of high confidence mutations detected by exome sequencing and targeted re-sequencing on a single colony level (**new figure 2D**).

2. We have performed single CFU-f colony re-sequencing from the initial MSC cultures, which were used for initiation of the exome sequenced cultures in our study for n=4 cases with n=33 colonies.

This experiment confirmed that high confidence mutations, which were detected in the expanded cultures were not detectable in clonally relevant frequency in the initiating material (**novel Table 3**).

In conclusion, both of these experiments fully confirmed our initial results and therefore add a third layer of validity to our data.

Reviewer #2 (Remarks to the Author): Expert in bone marrow microenvironment, stem cells and MSCs

The manuscript by Jann J-C et al is an interesting study on the possible meaning of mutations found in mesenchymal stromal cells (MSCs) from MDS patients, compared with healthy controls. The authors perform whole exome sequencing of expanded MSCs and targeted mutation analysis of primary samples. They confirm the presence of mutations in cultured MSCs previously reported in other studies. They discover multiple recurrent mutations but are unable to find most of these in the primary samples, although the number of cells available might be limiting. The authors conclude that the discovered mutations are culture artefacts and that there is no evidence for clonal mutations in the BM stroma of MDS patients. However, there are some technical limitations in the study, a wide age variability does not appear to be considered, the potential functional implications of recurrent mutations in MSCs or their progeny is not tested and the fact that the mutations are recurrent, some of them affect important pathways in MSC and bone-forming cells (such as RANK), should be considered as a note of caution when interpreting the results.

Specific comments:

1. The number of mutations is known to increase with age. The cohorts of MDS and healthy individual have 6-year difference in median age and a wide variability in age (36-86) which does not seem to be accounted for. Age should be included as an important factor in the equation.

Answer: We agree that the possible confounding factor age had not been completely accounted for in our data. We have therefore enlarged the exome sequencing data of our healthy control cohort by additional n=8 aged healthy individuals, now adding up to a total of n=28. By this adjustment, the age distribution between MDS and healthy cohorts are now fully balanced (**revised table 1, revised figure 1F and H**). Furthermore, we have added additional data depicting the number of detected mutations in dependency of age (**novel figure 1G**). Importantly, this improvement of age distribution between the cohorts did not change the results of our initial findings but rather more robustly confirms the asserted difference in the frequency of mutations in MDS MSCs versus healthy MSCs.

2. There is an inherent technical limitation in the ability to detect mutations dependent upon the number of cells (could this be measured?) and this might explain differences between primary cells and expanded cells needs to be taken into account both as a limitation of the study and in the interpretation of the results.

Answer: We fully agree with the reviewer that technical challenges of sequencing DNA from low cell numbers have to be taken into account for the analyses performed in this study and we have done so carefully: We have previously performed dilution and spike-in experiments to test and validate how low we can go down with the cell numbers and VAFs with our NGS protocols. These results were previously published in Mossner M et al. Blood 2016 (**supporting Figure for reviewers only from this publication**). Therein we conducted a dilution series of primary mutated and wild type cells and subjected decreasing amounts of cells to DNA extraction and subsequent identical amplicon based sequencing as performed in the current study. Thereby we could show, that for three independent

mutations, we were able to quantify mutational burden with VAFs <10% in samples with as little as 125 cells. For the FACS sorted samples in **Figure 5**, the median number of cells in the most stringent subfraction (Sytox-, CD45-, CD235a-, CD31-, CD271+) was 2928 cells per patient (range 68-74698). This range indicates, that two samples were sequenced from a cell number below our valid detection limit of 125 cells. Interestingly this was UPN1999, which was the only sample, in which a low level ZFX mutation was detectable in the stringent Sytox-, CD45-, CD235a-, CD31-, CD271+ fraction. The other sample was UPN252. We have therefore removed this data from these figures (**amended Figures 5D and 5F**). The underlying cell number range in the revised version is now (range 471-74698). In the less stringent subfraction (Sytox-, CD45-, CD235a-, CD271+/-, CD31+/-) cell numbers were considerably higher throughout. Thus, we conclude that the mutational calling from FACS sorted cells in this study is technically valid. We added this point to the discussion with reference to our previous publication in Blood 2016.

In Figure 6E, although independent samples of the same patients show that 4 out of 5 patients cluster together in the two samples, there is significant variability in the mutations detected for the same patient...does this variability also affects the recurrent mutations found?

Answer: We thank the reviewer for giving us the chance to present this data more clearly. As already outlined in the original manuscript version, mutational sets detected in these analyses were completely mutually exclusive. Unfortunately, none of these 5 patients carried any ZFX or RANK mutations. However, to address the reviewers question, we checked whether any of these samples contained any other recurrent mutations (e.g. listed in Table 2) and can report, that UPN462 carried a TTN and COL4A2 mutation in only one sample, that were not present in the corresponding matched paired sample. The same was true for mutations in DCH2 and LRP1B in UPN1002 and another TTN mutation in UPN1301. We have added these results to the manuscript to present this data more precisely.

3. The authors use the corresponding hematopoietic fraction as a germline control for the MSC exome sequencing data. However this reduces coverage since the authors have to exclude by hematopoietic affected genomic regions from analysis in the respective samples on an individual basis. How large are the genomic areas excluded and are there no mutations at all present in MSCs in these areas?

Answer: We thank the reviewer for addressing this important issue because we pointed it out in the original manuscript version but did not quantitatively specify the extent of the necessary curation of covered regions. We have now done this on a per patient level in the revised version of the manuscript. For any given patient, the deleted regions (e.g. on 5q) as determined by sequenza were removed from the calling region of the paired MSC samples. All removed regions are now summarized in novel **Supplemental Table 1**. Thereby 52/98 samples did not show any large scale LOH region in the bone marrow fraction and thus no genetic region was removed from calling at all. For the remaining minority of samples on average 9 Mb were removed per sample. This corresponds to a removal of an average of 1% per sample of the covered hg19 regions for the complete study cohort. We have added this additional information to the paper in the results section.

Is it possible that excluding different genomic areas in different patients (according to their mutational status) contributes to the observed heterogeneity?

Answer: We don't expect this blanking of LOH regions to be a major factor contributing to the observed heterogeneity because as outlined above, it represents only a very small fraction (approximately 1%) of the overall genomic regions analyzed. The most frequently removed regions were on chr 5q (24/92 removed sites) and chr 7q (11/92 removed sites). As outlined above, we have acknowledged this point in the discussion.

A more traditional germline control (not bearing known oncogenic drivers) might overcome these potential issues or limitations.

Answer: We have also addressed this point of alternative germline controls for further validation by performing exemplary exome sequencing of alternative germline samples such as CD3 sorted T-cells from peripheral blood and buccal swab DNA from the same patient (**novel Figure 2B**). In this exemplary case, we discovered a total of 40 high confident mutational calls when applying the same parameters as in Figure 1C. Thereby the great majority (38/40, 98%) were uniformly called with all different germline controls. Only 1 call (2%) was possibly missed by our strategy using bone marrow MNCs as germline control. In synopsis of the above described limited loss of coverage due to LOH region removal and the exemplary exome sequencing comparison of BM MNCs versus CD3+ T-cells versus buccal swab DNA we conclude that our chosen germline comparison is valid.

4. The correlation between the number of mutation and the duration of in vitro expansion (Figure 3C) does not seem as clear as claimed ($r=0.42$)

Answer: We agree that $r=0.42$ is not highly specific and have therefore toned down the respective text passage. However, we would like to rebut that this relative unspecificity is probably a consequence of the highly heterogeneous cell populations from primary patient material, which has been acknowledged multiple times in the manuscript and which is to be expected in primary human samples. The data were nevertheless clear and highly significant due to the unsurpassed high sample number of primary human MDS derived MSCs in this study.

5. P. 13. Related to “We therefore hypothesize that while previous mechanistic experiments in murine models have shown proof of concept that ectopic molecular alterations of the BM niche can induce MDS like phenotypes, this is not a relevant pathomechanism in adult human MDS.” This seems an overstatement... As the authors mention, although no clonal mutations were detectable in this study, other molecular alterations/signatures may in the MDS BM stroma may be pathogenic in MDS... Otherwise why would MSCs support and increase MDS HSC engraftment in previous studies? How to explain a higher mutational susceptibility of MDS MSCs? Additionally, there are germline mutations e.g. SBDS, PTPN11 associated with MDS/JMML development and their presence in the stroma has been shown to contribute to disease pathogenesis.

Answer: We thank the reviewer for this suggestion for improvement and have adapted the corresponding text passage as follows: “Since previous mechanistic experiments in murine models had shown proof of concept that ectopic molecular alterations of the BM niche could induce MDS like phenotypes, we also interrogated, whether we could detect stromal alterations known to induce MDS in murine models such as PTPN11, β -catenin, DICER1, SBD1 or RAR γ 61 in MDS MSCs but found no relevant mutations in these genes in our data.

6. P. 13. Related to “Therefore, we hypothesized that these events were largely secondary culture artefacts rather than true driver lesions originating from clonal mutations in the non-hematopoietic BM compartment” and “Discovery of valid mutations”. In the absence of functional studies assessing the impact of these recurrent mutations it is hard to judge whether they are functionally relevant or not. For instance, functional mutations that affect RANK might have important consequences in how these cells proliferate or differentiate in osteoblastic cells, and thereby influence their support of malignant hematopoietic cells. Although the authors claim that “even the high confidence mutations detected in serial MSC culture samples rather reflect outgrowth of particularly fit clones originating

probably of single cells selected by the culture condition”, the fact that these mutations are recurrent or selected in culture suggests a biological role that might secondarily affect the function of the malignant hematopoietic cells.

Answer: We agree with this discussion, especially the statements that the impact of recurrent mutations in MSC cultures is hard to judge without functional experiments and that they might possibly impact hematopoietic cells, e.g. in co-culture experiments. We have therefore performed some additional functional experiments with RANK mutated MDS MSCs versus RANK wild type MDS MSCs (**novel figure 3P, Q**) and showed proof of principle differential osteogenic differentiation dynamics. However, experiments of this kind with primary patient derived MSCs are extremely heterogeneous and difficult to control. According to our data, mutational profiles and VAFs in any culture of MDS MSCs used for functional experiments would not be predictable and they would be underlying highly dynamic changes of profiles and VAFs, which would have to be controlled throughout the course of e.g. a co-culture experiment. Therefore, functional experiments with BM MSCs should preferably be performed with the earliest possible passages. For the experimental data shown within this revision, we were able to re-thaw n=2 known RANK mutated MDS MSC cultures and confirmed the RANK mutations again for the functional experiment. However, for prospective experiments, probably only targeted genomically edited cells would be feasible, which goes beyond the scope of the current study and is somewhat theoretical since we cannot backtrack such mutations in vivo in MDS.

7. The different COSMIC signatures and their potential meaning need to be explained.

Answer: As requested we have added more specifically explanations. (Page 10)

8. What are the characteristics of UPN1999 that make explain this case standing out (age, mutational status, therapy etc)?

Answer: UPN1999 (Figure 5D) was a case of a 85 year old man, diagnosed with high risk MDS, who was referred to our center for second opinion after resistance to 5-azacitidine. This was the only case that showed low level (approx. 4% VAF, n=306 supporting reads/n=7983 total reads in the re-sequencing assay) detectability of a ZFX mutation in FACS purified non hematopoietic cells. As outlined above (point 2) this case had the lowest cell number (n=68) of sorted Sytox-, CD45-, CD235a-, CD31-, CD271+ cells, which formally was below our previously validated VAF quantification limit of n=125. We have therefore removed this data point along with the only other case, which had cell numbers below 125. In DNA from such low cell numbers, we hypothesize that fidelity during whole genome amplification is also reduced, which may have led to this result.

Reviewer #3 (Remarks to the Author): Expert in MDS genomics

In their manuscript, Jann, et al. investigate the possibility of clonal expansion in bone marrow environment in MDS patients, a long-standing issue in the field of MDS on the basis of several observations in human and in mouse. The authors analyzed in vitro-cultured BM-derived MSCs from both normal individuals (n=20) and patients with MDS (n=98) using whole exome sequencing to detect somatic mutations using BM hematopoietic cells as control. Serially obtained samples over 1-3 years were analyzed in 9 cases. The presence of detected mutations in in vitro cultured cells were tested in primary samples. The authors detected somatic mutations in samples from MDS patients

and normal individuals with unknown frequencies. Several genes, including ZFX, RNAK, TNFRSF11A, HTRA2, and PBM44, were significantly recurrently mutated. All mutations are found in a small subset of cultured cells with significantly larger VAFs in MDS-derived samples. However, most of these mutations were not stably detected during in vitro passages or detected in primary samples. On the basis of these observations, the authors concluded that these mutations are related to in vitro cell culture but does not represent clones positively selected in vivo. Although the conclusion is negative one, these observations are of interest worthwhile reporting. Meanwhile, several issues need to be addressed before it is considered for publications.

Major issues:

1) It should be clearly stated how many MDS and normal MSC samples had somatic mutations? Authors just reported distributions of VAFs.

Answer: We apologize that this had not become clear in the original version of the manuscript. As requested, we have specifically added numerical data on a per case basis in new **Supplemental table 1**, which supplements the data which was already shown in the original version (old figure 3A - now Figure 1F – depicting the frequency of mutations in MDS MSCs versus healthy MSCs). For better understanding we have also added an additional paragraph to more clearly describe this data (page 8).

2) Mutation calls in whole exome sequencing need to be validated at least for a subset of mutations for varying VAFs. This is critical.

Answer: We are thankful for pointing this out and apologize that this it did not become clear in the original manuscript. Validation of high confidence mutations in the MSCs cultures had actually already been carried out by amplicon based re-sequencing (old figures 5B-J). However, we fully agree that this was not clearly specified in the methods or results sections. Since other validation issues were also brought forward by other reviewers, we have accommodated this critique by implementing a complete new figure on validation experiments (**novel Figure 2**). Thereby we have also expanded on the initial validation experiments of our exome data by re-sequencing an additional n=120 mutations by targeted deep resequencing for this revision (**novel Figure 2A**). Furthermore, we have added an elaborate and detailed supplemental methods section, describing our NGS strategies in greater detail. With regard to sequencing fidelity in DNA from low cell numbers, we would also like to offer the appended “Reviewer only figure” from a previous study (Mossner M et al. Blood 2016 doi: 10.1182/blood-2015-11-679167), where we validated robust mutational and VAF quantification in low cell number samples with our applied protocol. As requested by other reviewers, we also implemented a third line of validation by also re-sequencing candidate mutations in single CFU-fs.

Importantly, in all validation experiments, we confirmed our original results.

3) As the authors mentioned, some control samples (i.e., MDS cells) were expected to have many copy number abnormalities, particularly LOH and should have complicated the detection of MSC-specific alterations. However, no details of the mutation calling methods were described. Please clarify how to detect MSC-specific mutations.

Answer: As outlined above, we apologize for not submitting this detailed information in the first submission and have now provided a detailed supplemental method section that more clearly states the mutational calling algorithms (**Supplemental Methods**). Further, as also pointed out to Reviewer 2, we have addressed the aspect of LOH specifically enumerating the extent of the copy number associated curation of the data and found that this had marginal effects on the total covered regions (mean total 1% of hg19 regions removed in the complete cohort).

4) Were there any mutational signatures that suggested an influence from in vitro culture?

Answer: A recent study by Kuijk et al Nat Com. 2020 also assessed the mutational impact of culturing human pluripotent and adult stem cells and likewise found that “in vivo mutational processes only play minor roles in vitro, where other mutational processes are dominant.” This is in line with our finding. By studying liver and intestinal stem cells they found in vitro-associated COSMIC Signatures 8 and 18 most dominant in in-vitro expanded pluripotent cells and argue that this correlated with varying oxygen levels. Given that healthy MSCs acquire different mutational signatures than MDS MSCs, we hypothesize that disease specific conditions (e.g. senescence) further mediate mutational acquisition during the in-vitro expansion. We have added a dedicated section addressing this point to the discussion (page 15).

5) Fig. 6A: it is problematic to decompose a small number of mutations in many cases, which may not represent real mutational processes. Thus, this may be misleading. So cases having small numbers of mutations, particularly <20 should be deleted.

Answer: We thank the reviewer for this justified suggestion and have amended the corresponding figure (now **new figure 3M**) to only include samples with >20 mutations. Expectedly, n= 30 samples were lost for clustering due to lower number of mutational calls and therefore clustering became less clear. Notably, the prediction of samples origin was still clearly possible with this higher stringency cutoff, with an even improved performance. The discrimination for del5q MSCs lost significance and was therefore removed from the manuscript.

6) Although recurrent mutations identified in in vitro-cultured cells were not confirmed in primary MSCs, it might be informative for authors to clarify the limitation in sensitivity in their study. In other words, to what extent can the authors exclude the possibility of the presence of such mutations.

Answer: This point was also raised by reviewer #2. Therefore, as already outline above, we are supplying validation data in a “reviewer only figure” from our previous study (Mossner M et al. Blood 2016 doi: 10.1182/blood-2015-11-679167). Therein we conducted a dilution series of primary mutated and wild type cells and subjected decreasing amounts of cells to DNA extraction and subsequent identical amplicon based sequencing as performed in the current study. Thereby we could show, that for three independent mutations, we were able to quantify mutational burden with VAFs <10% in samples with as little as 125 cells. In the present study, we applied the same methodology. For the FACS sorted samples in **Figure 5**, the median number of cells in the most stringent subfraction (Sytox-, CD45-, CD235a-, CD31-, CD271+) was 2928 cells per patient (range 68-74698). This range indicates, that two samples were sequenced from a cell number below our valid detection limit of 125 cells. Interestingly this was UPN1999, which was the only sample, in which a low level ZFX mutation was detectable in the stringent Sytox-, CD45-, CD235a-, CD31-, CD271+ fraction. The other sample was UPN252. We have therefore removed this data from these figures (**amended Figures 5D and 5F**). The underlying cell number range in the revised version is now (range 471-74698). In the less stringent subfraction (Sytox-, CD45-, CD235a-, CD271+/-, CD31+/-) cell numbers were considerably higher throughout. Thus, we conclude that the mutational calling from FACS sorted cells in this study is technically valid. We added this point to the discussion with reference to our previous publication in Blood 2016.

Minor issues:

7) Meaning of sentences is unclear here and there. Followings are just among many examples:

- P9: Vice versa, we correlated ...

- P9: These mutations most likely resembled...

- P9: ...calculated distinct clones...: clones cannot be calculated.

- P9: the last sentence.

- Many clumsy wordings and expression.

8) The use of 'subclonal ' is not appropriate, because the authors are using this to indicate that mutations are found in only in a subset of cells and there existed no 'major clones'. Please rephrase this wording.

9) English needs to be checked by a native speaker.

These issues have been addressed.

REVIEWERS' COMMENTS

Reviewer #1 (Remarks to the Author):

The authors have extensively revised the manuscript and added important data sets which support their initial findings and support the interpretation of the data. From my point of view I agree that the authors do not have to adjust the title. I do not have further comments or questions.

Reviewer #2 (Remarks to the Author):

Thanks to all authors for carefully addressing all of the issues raised. The revised article convincingly reports the absence of oncogenic drivers or selected clonal mutations in primary MSCs from MDS, which should be of great interest for the broad readership of Nat Commun. Instead, the data add to previous studies pointing towards the role of secondary functional alterations in MSCs as potential contributors to MDS pathogenesis.

Reviewer #3 (Remarks to the Author):

The authors addressed all the concerns this reviewer raised. I have no further concerns.